# MarR Family Transcriptional Regulators and Their Roles in Plant-Interacting Bacteria

**DOI:** 10.3390/microorganisms11081936

**Published:** 2023-07-29

**Authors:** Fanny Nazaret, Geneviève Alloing, Karine Mandon, Pierre Frendo

**Affiliations:** Université Côte d’Azur, INRAE, CNRS, ISA, 06903 Sophia Antipolis, France; fanny.nazaret@inrae.fr (F.N.); genevieve.alloing@unice.fr (G.A.); karine.mandon@univ-cotedazur.fr (K.M.)

**Keywords:** MarR, transcriptional regulation, signaling pathways, plant-interacting bacteria, phenolic compounds

## Abstract

The relationship between plants and associated soil microorganisms plays a major role in ecosystem functioning. Plant–bacteria interactions involve complex signaling pathways regulating various processes required by bacteria to adapt to their fluctuating environment. The establishment and maintenance of these interactions rely on the ability of the bacteria to sense and respond to biotic and abiotic environmental signals. In this context, MarR family transcriptional regulators can use these signals for transcriptional regulation, which is required to establish adapted responses. MarR-like transcriptional regulators are essential for the regulation of the specialized functions involved in plant–bacteria interactions in response to a wide range of molecules associated with the plant host. The conversion of environmental signals into changes in bacterial physiology and behavior allows the bacteria to colonize the plant and ensure a successful interaction. This review focuses on the mechanisms of plant-signal perception by MarR-like regulators, namely how they (i) allow bacteria to cope with the rhizosphere and plant endosphere, (ii) regulate the beneficial functions of Plant-Growth-Promoting Bacteria and (iii) regulate the virulence of phytopathogenic bacteria.

## 1. Introduction

Soil microorganisms need to adapt to numerous changes in their environment. These modifications can be associated with abiotic stresses such as an elevation in temperature [1,2], metallic contamination [3] or pollution by chemicals [4]. Soil microorganisms also have to deal with the plant rhizosphere, which is a selective ecological environment [5,6]. Therefore, they may benefit from the plant endosphere, which provides more nutrients and protection compared to the soil or rhizosphere [7]. Plant-associated microbial communities, either endophytic or associative, could play differential roles that are relevant to plant development [8]. The microorganisms encountered by plants form commensal, beneficial or pathogenic associations [9]. Among the beneficial microorganisms, Plant-Growth-Promoting Bacteria (PGPB) enhance plant growth via multiple mechanisms, including phytohormone production, biological nitrogen (N_2_) fixation (BNF) and the activation of plant defense [10]. Endophytic microorganisms also include phytopathogenic bacteria that significantly impair plant development [11]. To ensure a successful interaction, all of these plant-interacting bacteria must constantly adapt to the complex and changing phytosphere environment [12].

Bacteria developed a diverse set of complex regulatory pathways, resulting in a wide phenotypic plasticity. These pathways involve environmental sensor proteins that allow bacteria to detect and respond appropriately to fluctuating environments. Among them, transcriptional regulators differentially bind specific DNA sequences in the promoters of targeted genes depending on the signal perception. In this way, they promote (as activators) or prevent (as repressors) transcription initiation by RNA polymerase. MarR (Multiple antibiotic resistance Regulator) family regulators—whose first *marR* gene was identified in *Escherichia coli* [13,14]—generally act as repressors. However, some of them can also be activators, or sometimes exhibit both transcriptional repression/activation activities [15,16,17]. Their activities are regulated by phenolic (e.g., salicylate), metabolic (e.g., p-coumarate), metallic (copper, zinc) or toxic compounds (antibiotics, oxidants), which generally leads to the release of the transcriptional repression of the targeted genes. Most MarR family regulators form dimers containing a winged Helix-Turn-Helix (wHTH) DNA-binding domain flanked by helices involved in protein dimerization (Figure 1). MarR-encoding genes are widespread in over 12,000 bacteria and archaea genomes [18], and the average number of MarR proteins encoded by bacterial genomes varies greatly from one species to another [19,20]. Although MarR family transcriptional regulators were originally described as regulators of very small regulons, recent studies highlighted that some of them can also function as global regulators, differentially regulating the genes involved in various signaling pathways [21,22]. Despite their ubiquitous presence in bacteria, *marR* sequences remain poorly conserved between lineages, with generally less than 30% identity. This variability would explain the ability of some MarR-like regulators to interact with a variety of target promoters and to respond to a diverse range of physiological and environmental signals [20].

MarR-like regulators usually bind 16–20 bp palindromic sequences [23,24]. Some MarR-like regulators can also bind longer sequences, such as the *Dickeya dadantii* MarR regulator MfbR, which binds a single site of 48 bp, suggesting that another protein interacts with MfbR to modulate the target gene expression [25,26]. The DNA-binding activities of MarR-like regulators are regulated by a mechanism of allosteric modulation, induced by a specific ligand or small signal molecules [27,28]. The nature of the perceived signal (phenolic compounds, oxidants, etc.) depends on the ecological niche colonized by the bacteria, which allows for the activation of specific signaling pathways.

Several MarR-like regulators have been identified in plant-interacting bacteria, and they regulate numerous biological processes, including oxidative stress resistance and phytopathogens virulence. The present study provides an overview of the MarR family regulators in these bacteria and whether their interactions with plants are beneficial or pathogenic. The mechanisms regulating the activity of the MarR-like transcriptional regulators [24,29,30] will be described in the first part of the manuscript, and then the physiological roles of these regulators in rhizospheric and endophytic bacteria will be detailed. 

## 2. Regulation of MarR-like Transcriptional Regulators’ Activity by Environmental Chemical Signals

The activity of MarR family regulators depends on their ability to detect the chemical signals associated with the bacterial environment and use them for the transcriptional regulation of gene expression (Figure 2). MarR family regulators can be classified depending on the nature of the perceived signal. Some of these regulators can bind toxic compounds like antibiotics [29] or plant-derived metabolic compounds such as phenolic molecules (salicylate, protocatechuate, p-coumarate) [18,30,31,32]. Other MarR-like ligands are nitrogenous derivatives: for instance, the MarR/UrtR transcriptional regulators are able to bind urate and constitute a specific subfamily [27,33]. Moreover, some MarR-like regulators can bind metals such as copper, zinc and iron [34,35]. Finally, they may also contain one or two redox-sensitive cysteine residues that allow for the perception of oxidizing molecules [36,37,38]. In this context, the identification of the perceived signal by MarR-like regulators is a crucial step for their characterization, but this has succeeded in few cases [27]. As an example, salicylate was first considered as the main ligand of the *E. coli* MarR regulator, but crystallographic studies showed that copper, released during environmental stress or after an antibiotic treatment, is the natural inducer of *E. coli* MarR [39].

Crystallographic studies on MarR-like proteins bound to inducing ligands and coupled with amino-specific mutagenesis allow for the characterization of several ligand-binding mechanisms [25]. Generally, ligand molecules bind to MarR-like regulators at the interface between dimerization and DNA-binding domains, such as the protocatechuate ligand of the *Streptomyces coelicolor* PcaV regulator [18]. Interestingly, the number of ligand molecules binding to MarR-like regulators varies from one regulator to another. For example, *E. coli* and *Methanobacterium thermotrophicum* MarR-like regulators are able to bind one or two molecules of salicylate per monomer [23,40], while the *Staphylococcus epidermidis* TcaR regulator binds up to eight salicylate molecules per dimer [41]. MarR-like regulators could bind their ligand through different mechanisms. The superposition of MarR-ligand structures based on eleven known regulators (seven binding salicylates, two protocatechuates, one ethidium bromide and one 4-hydroxyphenylacetate) reveals a conserved binding pocket in the regulators [25]. Ligand binding usually induces a conformational switch, reducing the DNA-binding affinity of the wHTH motif. For example, p-coumaroyl-CoA binding to the *Rhodococcus jostii* CouR regulator involves a phenolic moiety inserted in a deep hydrophobic binding pocket [42]. In contrast, the *Pseudomonas aeruginosa* MexR is inactivated via an allosteric interaction mediated by the binding of an antirepressor peptide named ArmR [43]. However, this reduction in affinity is not always due to an allosteric modification. For instance, it has been proposed that ligand binding involves a Coenzyme-A ligand extension that occludes the CouR DNA-binding domain and impairs its binding to targeted gene promoters without inducing a conformational change [42].

Among the molecules capable of modifying protein structures, Reactive Oxygen Species (ROS) and Reactive Electrophile Species (RES) play an important role in the regulation of the bacterial signaling pathways. ROS are radical or nonradical oxidizing molecules generated during aerobic cell respiration as side products of the electron transport chain or tricarboxylic acid cycle (TCA) [44]. Proteins containing a thiol group exposed to solvents are particularly sensitive to redox-active compounds and oxidants such as hydrogen peroxide (H_2_O_2_). Depending on the environmental context, H_2_O_2_ exposure may lead to the formation of a reactive sulfenic acid (-SOH), which can further interact with another thiol group (-SH) to form a disulfide bond (S-S). Sulfenic acid could also be stabilized by interacting with other chemical groups (alkylation, nitrosylation) or with low-molecular-weight thiol compounds such as glutathione (GSH) [45]. Many MarR-like regulator activities are regulated in response to oxidants. The mechanism of organic hydroperoxides sensing used by the MarR/OhrR subfamily is based on one or two redox-active cysteine residues, which are involved in the formation of disulfide bonds or sulfenic acid derivatives [36]. Regulators belonging to the MarR/OhrR subfamily have been divided into two groups based on the number of redox-active cysteine residues. The *Bacillus subtilis* OhrR regulator (OhrR_Bs_) is a 1-Cys type, with a single redox-active cysteine residue at the N-terminus of the protein. In contrast, the *Xanthomonas campestris* OhrR regulator (OhrR_Xc_) is a 2-Cys type that harbors two redox-sensitive cysteine residues [37,46]. In the presence of organic hydroperoxides, the oxidation of a cysteine residue leads to the formation of sulfenic acid, but this sulfenylation is not sufficient to induce the derepression of the targeted gene [36] (Figure 2B). As a result, the oxidation of the unique OhrR_Bs_ C15 cysteine residue in sulfenic acid allows for the formation of a mixed disulfide bond with a low-molecular-weight intracellular thiol named bacillithiol (BSH, an antioxidant similar to GSH), leading to the dissociation of the regulator–DNA interaction [37]. In *X. campestris*, the oxidation of the OhrR_Xc_ C22 residue promotes the formation of an intermolecular disulfide bond with the C127 residue of the opposite monomer [47]. This intermolecular bond induces a major conformational change and brings cysteine residues in close proximity, resulting in the rotation of the OhrR_Xc_ wHTH motif, which fails to bind the DNA [36]. Meanwhile, the MarR/DUF24 subfamily includes RES sensor proteins. RES derivate from amino acids, lipids or products of carbohydrate oxidation, including quinones, diamides or aldehydes [36,48]. These molecules may also react with a redox-active cysteine residue and induce an interaction with low-molecular-weight compounds. For example, the *B. subtilis* YodB regulator, which controls the nitric oxide reduction and catechol degradation pathways, harbors three cysteine residues: C6, C101 and C108. In the presence of diamide, YodB can form an intermolecular disulfide bond between C6 and C101′, leading to a major conformational change in the DNA recognition helices [49]. However, the oxidized quinone provokes the S-alkylation of the C6 and C6′ cysteine residues and a reduction in the distance between the two wHTH motif helices. Then, the quinone-mediated S-alkylation significantly reduces the binding of YodB to its targeted sequences [50]. Finally, a few MarR family transcriptional regulators have been described to respond to metals through perception by redox-active cysteine residues. For example, Cu^2+^, released in cytosol upon membrane stress, leads to *E. coli* MarR and *Burkholderia thailandensis* BifR oxidation [39,51]. Indeed, Cu^2+^ oxidizes the C80 cysteine of *E. coli* MarR, leading to the formation of a disulfide bond between two dimers. The resulting tetrameric complex sequesters the DNA recognition helices, impairing DNA binding. The oxidation of BifR similarly leads to a tetramer formation but, instead of reducing the DNA binding, the oxidizing conditions result in the additional repression of the targeted genes by BifR, which functions as a “super-repressor” [39]. In conclusion, the regulation of the MarR transcriptional activity could involve both the interaction of MarR-like regulators with plant-deriving chemical compounds or redox-associated structural modifications of the regulator.

## 3. MarR Family Transcriptional Regulator’s Roles in Plant-Interacting Bacteria

Plant-interacting bacteria cope with numerous environmental changes, leading to the activation of various signaling pathways to adapt the cell responses. In this context, a wide variety of MarR-like regulators identified in plant-interacting bacteria are required for bacteria to adapt to their environments (Figure 3; Table 1). These regulators are involved in the regulation of several biological processes like (i) aromatic compound degradation in the plant rhizosphere, (ii) exopolysaccharide biosynthesis required for bacterial infection, (iii) oxidative stress resistance and (iv) phytopathogen virulence **[52]**.

### 3.1. Regulation of Aromatic Compounds Degradation in Plant Rhizosphere

Hydroxycinnamates (HCA) are plant secondary phenolic metabolites derived from the phenylpropanoid pathway. Among them, p-coumarate and ferulate are also found as Coenzyme-A thioesters (p-coumaroyl-CoA and feruloyl-CoA), which can be used for flavonoids or lignin biosynthesis [82]. A root exudation may provoke the release of HCA that are potentially toxic for bacteria in the rhizosphere [83]. However, some species are able to detoxify these compounds by enzymatic pathways and use them as a carbon source [31,84]. There are two type of HCA degradation pathways: the β-oxidative and the non-β-oxidative pathway, and both produce protocatechuate that is then degraded via the β-ketoadipate pathway before it enters the TCA cycle [85] (Figure 4). The two first β-oxidative and non-β-oxidative degradation steps are regulated by MarR family transcriptional regulators in numerous bacteria (Table 1). The p-coumarate β-oxidative degradation has been described in the rhizosphere bacteria *Rhodopseudomonas palustris*. Here, the first degradation steps involve the Coenzyme-A ligase encoded by *couB*, which produces p-coumaroyl-CoA, and an enoyl-CoA hydratase encoded by *couA*. The expression of these two genes is repressed by CouR, a MarR-like transcriptional regulator, whose activity is inhibited upon the binding of p-coumaroyl-CoA [31,86]. Consequently, CouR inactivation in an *R. palustris* mutant leads to a rapid bacterial adaptation during transfer into a medium containing p-coumarate as the sole carbon source [86]. In *Agrobacterium fabrum* C58, the β-oxidative degradation of HCA involves genes of the SpG8-1b cluster, whose expression is repressed by the MarR-like regulator named HcaR [87]. These genes are expressed thanks to the release of the HcaR DNA-binding activity in the presence of p-coumaroyl-CoA and feruloyl-CoA. HCA degradation confers a competitive advantage to *A. fabrum* C58. As a result of the *hcaR* deletion, a mutant constitutively expressing the genes of the SpG8-1b cluster shows a higher growth rate than the wild-type (WT) strain in the tomato plant rhizosphere [55]. In addition to its saprophytic lifestyle, *A. fabrum* C58 can induce tumor formation through its Ti plasmid carrying virulence genes [88]. Interestingly, an *hcaR* mutation leads to a decreased virulence and causes the *A. fabrum* C58 mutant to have a competitive disadvantage compared with the WT in the tumor. These observations suggest that HcaR activity could be required for the induction of virulence genes. Then, the regulation exerted by HcaR in response to HCA may promote the *A. fabrum* C58 transition between its saprophytic and phytopathogenic lifestyles [55].

In *Acinetobacter* sp. ADP1, HCA degradation is non-β-oxidative and depends on the activity of a regulator also named HcaR, which represses this pathway in the absence of Coenzyme-A esters [53,89]. The repressor inactivation promotes bacterial growth on a medium containing HCA as the only carbon source [53]. The first two steps of the non-β-oxidative pathway of ferulate degradation lead to the formation of vanillin, which is not produced during the β-oxidative pathway. Both steps are catalyzed by a Coenzyme-A ligase and an enoyl-CoA hydratase/lyase, which are regulated by an HcaR regulator [53,89]. Vanillin is then converted to vanillate, which is ultimately degraded to protocatechuate before entering the β-ketoadipate pathway and supplying *Acinetobacter* metabolism. *Pseudomonas fluorescens* BF13 also carries out non-β-oxidative HCA degradation, regulated by FerR, a MarR-like regulator capable of both activation and repression [72]. Another study also reported the role of the DesR regulator of *Sphingobium* sp. SYK-6 in vanillate and syringate root metabolite catabolism through the protocatechuate pathway. In this case, vanillate is *O*-demethylated in one step by LigM and syringate in three steps by DesA, LigM and DesB enzymes. Araki and colleagues showed that DesR is a transcriptional repressor that negatively regulates the expression of *ligM*, *desA* and *desR*. In the presence of vanillate or syringate, DesR DNA binding is inhibited, allowing *Sphingobium* to catabolize the root metabolites [32]. Finally, the MarR family regulator YetL of *B. subtilis* controls the expression of *yetM* by encoding a flavonoid-responsive monooxygenase enzyme. The binding of flavonoids to YetL results in the activation of the *yetM* expression, which then allows *B. subtilis* to counteract the flavonoid toxicity and to adapt to the rhizosphere environment [64]. To summarize, the MarR-like regulators involved in plant-specific compound degradation allow bacteria to adapt to the rhizosphere. It would be interesting to study the regulation of such degradation in other rhizospheric bacterial species with phytobeneficial properties. For example, this regulation could be important in *Azospirillum*, which is exposed to HCA derivatives (such as N-p-coumaroylputrescine) in the rice rhizosphere [90]. 

In addition to plant secondary metabolites, diverse MarR-like regulators are involved in the catabolism of auxin, a plant hormone that plays critical roles in plant developmental processes including cell division and root development. Auxin degradation operons are present across the bacterial tree of life in two distinct types on the basis of gene content and metabolic products: *iac*-like and *iad*-like types. An analysis of the structures of the MarR-like regulators encoded by representatives of each auxin degradation operon type established that each has distinct IAA-binding pockets. A comparison of the representative IAA-degrading strains from the diverse bacterial genera colonizing *Arabidopsis thaliana* showed that only strains containing *iad*-like auxin-degrading operons interfere with auxin signaling in a complex synthetic community context. In these communities, IadR MarR-like regulators allow the bacteria to rapidly catabolize auxins [91]. This suggests that *iad*-like operon-containing bacterial strains play a key ecological role in modulating auxins in the plant microbiome.

### 3.2. Regulation of Exopolysaccharide Biosynthesis

Exopolysaccharides (EPS) are biological polymers secreted by microorganisms to cope with harsh environmental conditions. EPS are the main components of the extracellular biofilm matrix protecting microbial cells from adverse factors such as temperature, pH, antibiotics and host immune defenses [92]. EPS are essential for the establishment and maintenance of the symbiotic interaction between *Sinorhizobium meliloti* and *Medicago* sp. [93,94,95]. This symbiosis involves a rhizobial host root infection at the level of an infection pocket, formed by root hair curling. The bacteria invade the plant via an infection thread (IT), resulting in the formation of a nodule in which the bacteria differentiate into nitrogen-fixing bacteroides [96]. 

*S. meliloti* produces two categories of Exopolysaccharides: succinoglycans (EPS-I) and galactoglucans (EPS-II). Enzymes of the EPS-I biosynthesis pathway are encoded by the genes of the *exo* cluster, whereas EPS-II biosynthesis involves the genes of the *exp* cluster, which are physically distinct from *exo* genes. EPS-I are particularly important for IT formation and nodule infection [97,98], while EPS-II are produced extensively in the soil. It is known that EPS-II are required for biofilm formation and root colonization by *S. meliloti* [94]. The regulation of EPS biosynthesis involves a complex transcriptional regulation network [78]. The transcriptional regulation of EPS-II biosynthesis genes is differentially controlled by the MarR-like regulator WggR (formerly ExpG) and two other regulators, PhoB and MucR. Interestingly, depending on both the protein–protein interaction and phosphate concentration, WggR is able to positively or negatively affect EPS-II biosynthesis [77], contrasting with the previously described MarR-like activities. Given its high complexity, the transcriptional regulation of *exp* genes is divided into three distinct regulatory levels. For ease of use, only *wgaA*, *wgdA* and *wgeA* regulation is described. (i) The first level involves PhoB and WggR, which act cooperatively to activate *wgaA*, *wgdA* and *wgeA* genes under phosphate starvation [77,99,100]. (ii) The second level involves MucR, repressing the expression of *wgaA*, *wgdA* and *wgeA* independently of the phosphate concentration [77]. Interestingly, in an *mucR* mutant, WggR harbors a repression activity, suggesting that WggR requires PhoB to act as a positive regulator of EPS-II biosynthesis genes. (iii) The third level of regulation involves the Sin quorum sensing of *S. meliloti* and the LuxR-like regulator ExpR [101]. In the absence of MucR and above a certain population density, ExpR activates the expression of EPS-II biosynthesis genes. When MucR is present, its repression is counteracted by ExpR only in the presence of WggR [101,102]. In this context, WggR acts as a general mediator by cooperating with either PhoB or ExpR to counteract the repressive activity of MucR. The MarR-like regulator WggR of *S. meliloti* is thus an essential determinant of symbiosis by promoting the early stages of infection and root colonization [102] (Figure 5). 

### 3.3. Regulation of Resistance to Oxidative Stress

The bacteria interacting with plants can be exposed to oxidizing molecules such as ROS or RES [36,103]. Plants are able to generate RES such as quinones or diamides and organic hydroperoxides such as lipid hydroperoxides (LOOH) in response to bacteria [104]. Moreover, RES can also be produced by other soil microorganisms such as fungi [105,106]. In plant-interacting bacteria, there are different enzymatic mechanisms for ROS and RES detoxification, including peroxidases, catalases, alkyl hydroperoxide reductases and azoreductases [61]. RES detoxification pathways particularly involve azoreductases (AzoR1, AzoR2), nitroreductases (YodC) or thiol-dependent dioxygenases (CatE, MhqA) catalyzing the reduction or cleavage of electrophilic compounds [49]. *B. subtilis*, used for its phytostimulant properties [107], can be exposed to numerous oxidative stresses, in particular to RES [65]. Regulators of the MarR/DUF24 subfamily YodB, CatR and MhqR of *B. subtilis* regulate detoxification pathways, allowing for RES resistance. Thus, mutants unable to express YodB, CatR or MhqR showed increased resistance to RES compounds, catechol and 2-methylhydroquinone (2-MHQ) [58,60,65]. *azoR1* is regulated by YodB, while *catE* is regulated by both YodB and CatR [58,65]. MhqR also regulates the expression of genes encoding dioxygenases such as MhqA and the azoreductase AzoR2 [60]. In addition, these regulators appear to be important for *B. subtilis* development in the rhizosphere of some plants. For example, allicin and diallyl-polysulfanes (DAS4) produced by garlic plants (*Allium sativum*) are electrophilic antimicrobial compounds that induce the expression of genes controlled by YodB and CatR. Indeed, the presence of allicin or DAS4 causes the S-thioallylation of the redox-active cysteines of these repressors, which leads to their inactivation and alleviates the repression [108].

Organic hydroperoxides like LOOH are found in plant tissues [109]. The oxidative burst produced by the plant in response to infection by a pathogen leads to a sharp increase in these LOOH, which are toxic for microorganisms [110,111]. The OhrR repressor involved in the resistance to these organic hydroperoxides was first identified in *X. campestris* pv. *phaseoli*. The OhrR repressor is inactivated in the presence of LOOH such as peroxidized linoleic acid or alkyl peroxides such as *tert*-butyl (tBOOH), allowing for the expression of the *ohr* gene encoding an alkyl peroxide reductase [46,81,112]. Ohr-type alkyl peroxide reductases are known to be specialized in the detoxification of long oxidized lipid chains like LOOH. Indeed, the Ohr enzyme of *X. campestris* pv. *phaseoli* is essential for the survival of bacteria during oxidative stress induced by the addition of tBOOH or LOOH [81,112]. These enzymes are found in many plant-interacting bacteria and may be important for bacterial adaptation during host colonization. For example, the *ohrA* gene encoding alkyl peroxide reductase and regulated by an OhrR homolog is found in plant pathogenic bacteria of the genus *Agrobacterium* and in *Dickeya zeae* [54]. In the latter case, OhrR also regulates crucial functions for *D. zeae* phytopathogenicity [69]. This shows the importance of organic oxidant detoxification mechanisms in plant-associated bacteria.

ROS production by plants was also intensively described in beneficial plant–bacteria interactions. The O_2_^•−^, H_2_O_2_ and NO production by the host plant is particularly well documented in Rhizobium–legume symbioses, in which these ROS/RNS are produced at different stages of nodule development [113,114]. Then, symbiotic bacteria need to tightly regulate redox homeostasis in order to properly colonize the plant host. For example, *S. meliloti* exerts an efficient antioxidant defense consisting of several H_2_O_2_ and O_2_^•−^ detoxification enzymes [115,116]. *S. meliloti* also contains an Ohr-type alkyl peroxide reductase, whose expression is regulated in the presence of organic peroxides by OhrR, a homolog to the previously described OhrR of *X. campestris* [76]. The *S. meliloti* Ohr enzyme is essential for its survival in the presence of alkyl peroxides (tBOOH) and is expressed during symbiosis, particularly in nitrogen-fixing bacteroides. This suggests a production of organic hydroperoxides by the plant and the role of Ohr proteins in their detoxification [76]. However, plants inoculated with an *ohr* mutant of *S. meliloti* do not present nodulation defects, while a single deletion of the *ohr* gene in *Azorhizobium caulinodans* leads to a significant decrease in nitrogen fixation [57], suggesting the existence of functional equivalents of the Ohr enzyme in *S. meliloti* such as the AhpC alkyl hydroperoxidase [76,117]. Nevertheless, a recent study suggests that triple deletion in paralogous *ohrR* genes in *S. meliloti* affects symbiotic performance during interactions with *Medicago sativa* [118]. 

### 3.4. Regulation of Virulence of Phytopathogenic Bacteria

Infection by phytopathogens requires several strategies so the bacteria can survive, multiply and colonize the plant. Among them, the production of virulence factors is essential for the success of infection by phytopathogenic bacteria. The genes encoding these factors are involved in particular in motility, adhesion, tissue degradation and resistance to plant defenses. Moreover, plant infection is a complex process that requires the temporal regulation of the virulence genes’ expression and is often linked to numerous transcriptional regulators. 

Virulence regulators are well described in *Dickeya dadantii*, a phytopathogenic bacterium responsible for soft rot in a wide range of plants, including the model plant *Arabidopsis thaliana* [119]. This disease is caused by the degradation of plant tissue by pectinolytic enzymes such as cellulase and pectate lyases [69,120]. Fine tuning the expression of the genes involved in the biosynthesis and secretion of these enzymes is required for the pathogenicity of *D. dadantii* [68]. This regulation involves PecS, a global regulator of *D. dadantii* virulence belonging to the MarR family. On the one hand, PecS regulates the expression of the genes involved in host colonization such as the synthesis and secretion of pectinolytic enzymes (*pel*, *celZ*, *out* genes), flagellar-mediated motility (*fli* genes) and quorum sensing (*vfm* genes). On the other hand, PecS also plays a role in tolerance to plant defenses by regulating the expression of an alkyl hydroperoxide reductase (*ahpC* gene) or indigoidin (*indA* gene, by encoding a blue pigment with antioxidant properties) [68,121,122]. PecS also regulates the synthesis of harpin (*hrpN* gene), which is a virulence factor able to induce the hypersensitive response (HR) in plants [123]. Virulence genes are repressed by PecS during the so-called asymptomatic phase, when *D. dadantii* colonizes plant tissue and the intercellular apoplast. The repression of virulence genes by PecS during the asymptomatic phase is required to avoid the early activation of plant defenses, which would lead to the elimination of the pathogen during the first infection phase [68]. Infection is followed by a symptomatic phase, during which virulence factors are expressed and where bacteria initiate tissue maceration. The signal perceived by PecS in planta that allows for this derepression has not yet been identified. However, in vitro studies showed that the PecS homolog of *Agrobacterium fabrum* is able to bind urate, which attenuates its binding to promoter regions [56]. Urate is produced by the action of xanthine oxidases involved in purine metabolism in many plants and has antioxidant properties [124]. Thus, it was proposed that urate can be used as a signal perceived in planta to regulate the expression of bacterial virulence genes via PecS [56]. Furthermore, the PecS homolog of *Pectobacterium atrosepticum*—also responsible for soft rot disease—responds in vitro to H_2_O_2_ and organic hydroperoxide treatments [71]. However, it should be noted that PecS is not the only regulator of *D. dadantii* virulence. In total, ten regulators are mainly involved, including the MarR-like regulators PecS, MfbR and SlyA [125]. On the one hand, MfbR induces the expression of genes encoding plant-wall-degradation enzymes (PCWDE). This activation is associated, directly or indirectly, with the alkalinization of the pH encountered at advanced stages of the infection [26]. On the other hand, SlyA is able to activate the expression of numerous virulence genes, in particular those encoding the production of pectate lyases or the formation of a type-III secretion system, which are useful for *D. dadantii* [16,17]. In *D. zeae*, SlyA also regulates numerous virulence factors, such as zeamines and phytotoxins, and is crucial to regulate plant colonization. Zhou and colleagues showed that *slyA* inactivation leads to a significant decrease in phytopathogenicity on rice and in biofilm formation, and an increase in swimming and swarming motility [70]. Interestingly, a study showed that *slyA* expression is positively regulated by the *D. zeae* OhrR regulator, suggesting a link between redox signaling pathways and the MarR family regulators involved in plant–bacteria interactions [69]. 

MarR-like regulators are also found in many other phytopathogenic bacteria (Table 1). For example, *Burkholderia* is a phenotypically diverse bacterial genus containing species that can cause many plant and animal diseases [126]. Interestingly, bacteria of the genus *Burkholderia* have a large number of MarR-like regulators that may participate in this diversity [66]. Among all the annotated MarR-like regulators in *Burkholderia*, four have been characterized: MftR, BifR, OhrR and TctR [15,51,67]. They play a role in host colonization by controlling the expression of virulence genes (MftR), biofilm production (BifR), resistance to organic hydroperoxides (OhrR) and the formation of a type-VI secretion system (TctR). In addition, the PrhN regulator of *Ralstonia solanacearum* also regulates the virulence of the bacteria, notably by activating the expression of an *hrp* gene cluster involved in the formation of a type-III secretion system used to inject virulence factors into host cells. The deletion of the *prhN* gene in *R. solanacearum* significantly reduces its virulence and the colonization of tomato plants [73]. Other MarR-like regulators are general regulators of virulence as described in *D. dadantii*; an example is the HpaR regulator of *X. campestris* pv. *campestris* [79]. A mutant deleted in the *hpaR* gene was nonvirulent and unable to trigger a hypersensitive response in the plant [80]. Furthermore, the HpaR regulator directly regulates the expression of the *vgrR-vgrS* operon encoding a two-component system involved in iron homeostasis, which is important for bacterial virulence [79,127]. 

## 4. Conclusions

Bacterial adaptation to their dynamically changing environments involves key regulatory mechanisms relying on transcriptional regulation. Whether bacteria are pathogenic, commensal or symbiotic, they must sense their host, evade the immune system, compete with other microbes for space and nutrients and adapt their physiology to the newly established niche. It is worth noting that the regulatory mechanisms underlying adaptation are shared between symbiotic and pathogenic bacteria. Here, we particularly reviewed the numerous roles of MarR family transcriptional regulators during plant–bacteria interactions, but MarR-like regulators are also crucial for clinically relevant pathogens. It is possible that regulators shared by pathogenic bacteria and symbiotic bacteria have played roles in the evolution of a pathogenic to a symbiotic interaction, or of a saprophytic to pathogenic interaction, as they can regulate gene expression in ways that are beneficial for both lifestyles. On the other hand, the presence of MarR family regulators in various bacterial species is not necessarily correlated with the conservation of perceived signals or regulated genes when they are known. Finally, MarR-like regulators are also crucial for adaptation to extreme abiotic conditions, such as those encountered by archaea.

Although this study highlights the importance of transcriptional regulation in bacterial adaptation, it should be pointed out that bacteria remain complex organisms. Because of this fundamental complexity, studies on transcriptional regulation mechanisms in plant–bacteria interactions alone are not sufficient to fully explain their functioning. Nevertheless, they provide the essential background for the development of integrative analyses. Then, by obtaining benefits from multidisciplinary approaches and from the mass implementation of bioinformatics and an OMICs analysis, challenging research in the field will provide new evidence concerning the evolution of the transcriptional regulatory circuits involved in bacterial adaptation to biotic and abiotic environments.

## Figures and Tables

**Figure 1 microorganisms-11-01936-f001:**
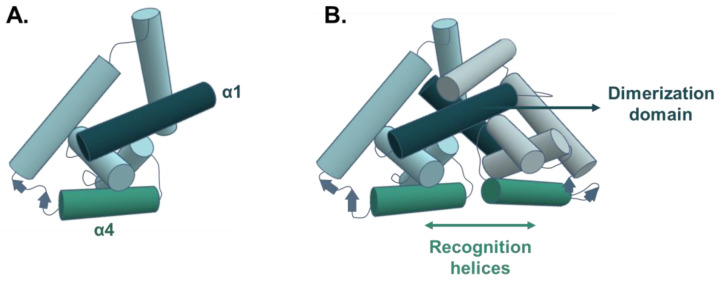
Schematic representation of MarR-like regulator’s structural features. (**A**). MarR-like monomer subunit with secondary structure elements. (**B**). MarR-like dimer organization. The two long intersecting helices α1 (dark blue arrow) form the dimerization domain of the two MarR-like monomers, and the two recognition helices α4 (green arrow) form the wHTH DNA-binding domain. The α2, α3, α5 and α6 helices are shown in light blue. Based on PDB ID: 3QPT.

**Figure 2 microorganisms-11-01936-f002:**
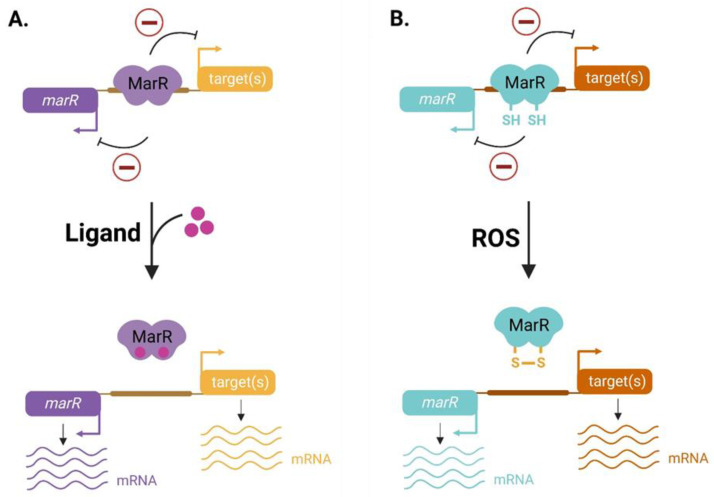
Regulation of MarR-like repression activity by environmental signals. (**A**) Regulation by ligand binding. MarR-like regulators generally bind ligand molecules at the interface between dimerization and DNA-binding domains and then experience a destabilizing interaction with DNA and induce expression of targeted genes. (**B**) Regulation by ROS oxidation. Some MarR-like regulators harbor cysteine residues able to perceive ROS signals. Oxidation of cysteine residues induces the formation of disulfide bond, modifying protein conformation and preventing repression of targeted genes.

**Figure 3 microorganisms-11-01936-f003:**
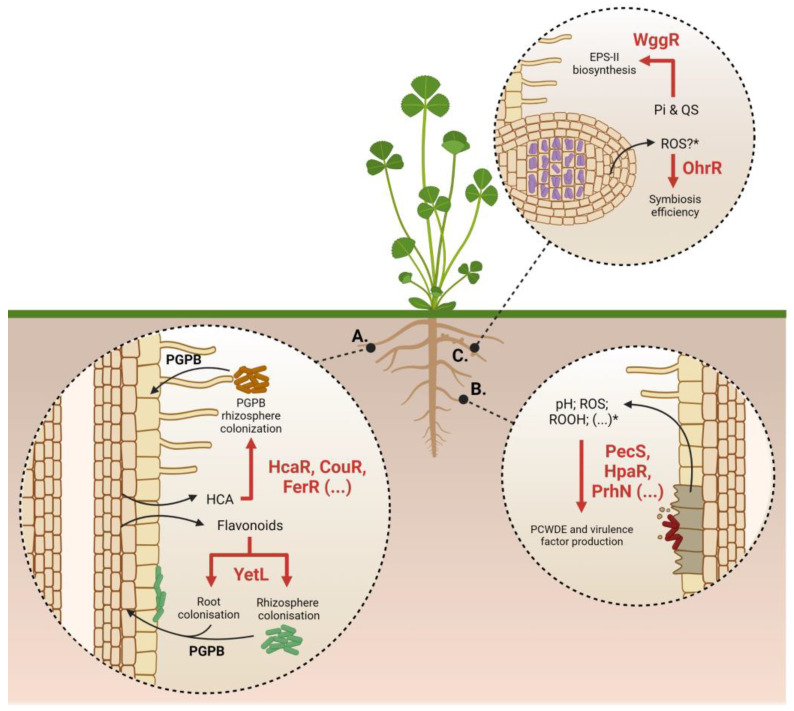
Model for MarR-like regulated pathways involved in plant–bacteria interactions. (**A**) Roles of MarR regulators in rhizospheric bacteria. (**B**) Roles of MarR regulators in phytopathogenic bacteria. (**C**) Roles of MarR regulators in symbiotic bacteria. *: No evidence concerning the plant signal recognized by the MarR-like regulator. PGPB: Plant-Growth-Promoting Bacteria; HCA: Hydroxycinnamates; EPS: Exopolysaccharides; Pi: Inorganic phosphate; ROS: Reactive Oxygen Species; ROOH: organic hydroperoxides; PCWDE: Plant-Cell-Wall-Degrading Enzymes.

**Figure 4 microorganisms-11-01936-f004:**
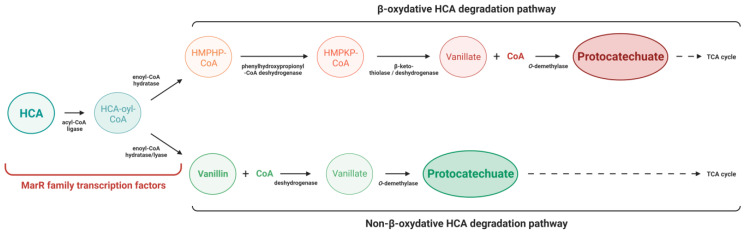
Degradation of HCA by β-oxidative and non-β-oxidative pathways. The first two steps of HCA degradation pathways are regulated by MarR-like regulators (red) in *Rhodopseudomonas palustris* and *Agrobacterium fabrum* C58 in the β-oxidative pathway and in *Acinetobacter* sp. ADP1, *Pseudomonas fluorescens* BF13 and *Sphingobium* sp. SYK-6 in the non-β-oxidative pathway. p-coumarate and ferulate are degraded by these pathways. The protocatechuate released at the end of the pathway is then incorporated into the TCA cycle via the β-ketoadipate pathway. HCA: Hydroxycinnamates; CoA: Coenzyme A; HMPHP: 4-hydroxy-3-methoxyphenyl-β-hydroxypropionic acid; HMPKP: 4-hydroxy-3-methoxyphenyl-β-ketopropionic acid [55].

**Figure 5 microorganisms-11-01936-f005:**
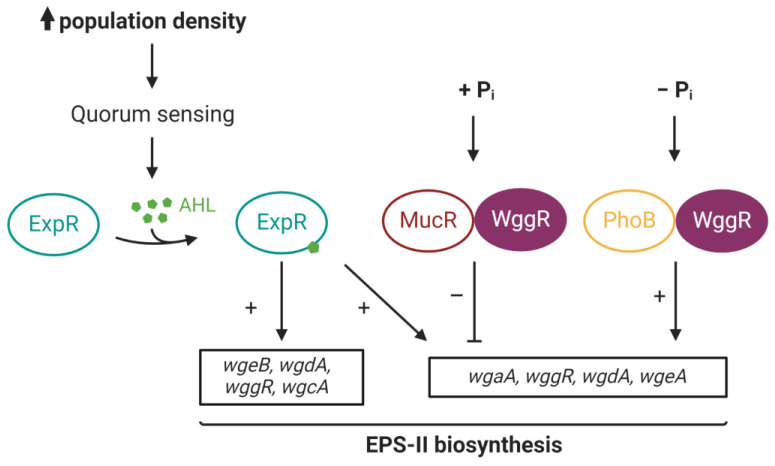
Regulation network model for EPS-II biosynthesis in *Sinorhizobium meliloti*. Under phosphate-starvation conditions, WggR (purple) and PhoB (orange) cooperatively induce EPS-II biosynthesis. At higher phosphate concentrations, only MucR (red) and WggR repress the expression of EPS-II biosynthesis genes. When the population density increases, AHL (green) produced by bacteria are recognized by the ExpR receptor regulator (light blue). ExpR can then induce the expression of EPS-II biosynthesis genes and counteract the repression exerted by MucR (in the presence of WggR). AHL: Acyl Homoserine Lactone; Pi: Inorganic phosphate. *wgeB*, *wgdA*, *wggR*, *wgcA*, *wgaA* and *wgeA* genes belong to the *exp* cluster [78].

**Table 1 microorganisms-11-01936-t001:** MarR family regulators characterized in plant-interacting bacteria. Regulated functions are defined according to the regulated gene or to an observed phenotype in rhizosphere or in planta. On average, bacteria harbor at least seven MarR-like paralogs per genome. Only characterized regulators are presented (nonexhaustive list).

Organism	MarR-like	Regulation	Signal	Regulated Functions	References
*Acinetobacter* sp. ADP1	HcaR	R	p-coumaroyl-CoA; feruloyl-CoA	HCA degradation	[53]
*Agrobacterium fabrum* C58	OhrR	R	ROOH	ROOH resistance	[54]
	HpaR	R	p-coumaroyl-CoA; feruloyl-CoA	HCA degradation	[55]
	PecS	R	Urate	ROS resistance	[56]
*Azorhizobium caulinodans*	OhrR	R/A	ROOH	ROOH resistance	[57]
*Bacillus subtilis*	CatR	R	RES; NaOCl	RES resistance	[58]
	HypR	A	RES; NaOCl	NaOCl resistance	[59]
	MhqR	R	RES	RES resistance	[60]
	OhrR	R	ROOH; NaOCl	ROOH resistance	[61,62]
	YetL	R	Flavonoids	Flavonoid detoxification	[63,64]
	YodB	R	RES; NaOCl	RES resistance	[58,65]
*Burkholderia* sp.	BifR	R	Copper	Biofilm formation	[51]
	MftR	R	Urate	Virulence	[15]
	OhrR	R	ROOH	ROOH resistance	[66]
	TctR	R/A	N.D	Virulence	[67]
*Dickeya dadantii*	MfbR	R/A	pH	PCWDE biosynthesis	[26]
	PecS	R	N.D	Virulence	[68]
	SlyA	R/A	N.D	Virulence	[16,17]
*Dickeya zeae*	OhrR	R/A	N.D	Virulence; ROS resistance	[69]
	SlyA	R/A	N.D	PCWDE biosynthesis	[70]
*Pectobacterium atrosepticum*	PecS	R	Urate; H_2_O_2_; ROOH; pH	ROS resistance	[71]
*Pseudomonas fluorescens*	FerR	R/A	Feruloyl-CoA	HCA degradation	[72]
*Ralstonia solanacearum*	PrhN	A	N.D	Virulence	[73]
*Rhodopseudomonas palustris*	BadR	R/A	2-ketocyclohexane-1-carboxyl-CoA	Benzoate degradation	[74]
	CouR	R	p-coumaroyl-CoA	HCA degradation	[31]
*Sphingobium* sp. SYK-6	DesR	R	Vanillate; syringate	Lignin degradation	[32]
	FerC	R	p-coumaroyl-CoA; feruloyl-CoA	HCA degradation	[75]
*Sinorhizobium meliloti*	OhrR	R	ROOH	ROOH resistance	[76]
	WggR	R/A	Phosphate concentration	EPS-II biosynthesis	[77,78]
*Xanthomonas campestris*	HpaR	R/A	N.D	Virulence	[79,80]
	OhrR	R	ROOH	ROOH resistance	[46,81]

R: repressor; A: activator; N.D: nondescribed; ROOH: organic hydroperoxides; ROS: Reactive Oxygen Species; RES: Reactive Electrophile Species; PCWDE: Plant-Cell-Wall-Degrading Enzymes.

## Data Availability

No new data were created or analyzed in this study. Data sharing is not applicable to this article.

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
