# Peer review of "MarR Family Transcriptional Regulators and Their Roles in Plant-Interacting Bacteria"

_microorganisms, 2023, doi:10.3390/microorganisms11081936_

Round 1

Reviewer 1 Report

Dear Athors,

I found the reviewed manuscript to be very well written. The content of the topic has been comprehensively described. 

Author Response

We thank reviewer 1 for his comments.

Reviewer 2 Report

The manuscript ID: microorganisms-2493299 entitled "MarR family transcriptional regulators and their roles in plant-2 interacting bacteria" by Nazaret et al. Overall, this is an interesting paper that provides promising data for an important issue in Plant Growth where the reported in this  review focuses on mechanisms of plant signal perception by MarR-like regulators i) allowing bacteria to  cope with rhizosphere and plant endosphere, ii) regulating the beneficial functions of Plant Growth Promoting Bacteria and iii) regulating the virulence of phytopathogenic bacteria. In my opinion, the minor strength of this review is the critical mechanistic discussion of the pervious reported used to prove the accuracy of this review aims. That said, there are also a few shortcomings of the article that I encourage to be explored further. The manuscript is need MINOR revision to be suitable to publish due to the following issues,

1-      The manuscript should be thoroughly reviewed with native English speakers, several sentences are not clear and do not provide any clear meaning.

2-      Line 75 Delete (….) and change it to (etc.)

3-      I recommend discussing limitations of the study in the discussion section.

4-      Suggest remove the references from the conclusion. 

Need minor editing of English 

Author Response

We thank reviewer 2 for his comments and suggestions. We have answered point per point to the comments/suggestions in bold.

1-      The manuscript should be thoroughly reviewed with native English speakers, several sentences are not clear and do not provide any clear meaning.

As the reviewer did not indicate which sentences are not clear or do not provide clear meaning, and that the three other reviewers find that the quality of English language is fine, we did not give the manuscript to review to a native English speaker.

2-      Line 76 Delete (….) and change it to (etc.)

The modifications were performed in the text (see edited text in red).

3-      I recommend discussing limitations of the study in the discussion section.

We have introduced a discussion about the limitations of the study in line 471 of the conclusion section (see edited text in red).

4-      Suggest remove the references from the conclusion. 

We have removed the references in conclusion section.

Reviewer 3 Report

The present study provides an overview of MarR family regulators in bacteria. MarR-like transcriptional regulators are essential for the regulation of specialized functions involved in plant-bacteria interactions in response to a wide range of molecules associated with the plant host. The review is interesting and meaningful. I have some suggestions.

1. What’s the difference between plant-bacteria interactions and plant-fungi interactions in the MarR family?

2. What’s the most important scientific gap of the MarR family regulators in bacteria?

It's fine.

Author Response

We thank reviewer 3 for his comments and suggestions. We have answered point per point to the comments/suggestions in bold.

  1. What’s the difference between plant-bacteria interactions and plant-fungi interactions in the MarR family?

To our knowledge, MarR regulators are restricted to bacteria and archea.

  1. What’s the most important scientific gap of the MarR family regulators in bacteria?

We have introduced a short discussion about the scientific gap to be filled in the conclusion (see edited text in red).

Reviewer 4 Report

The paper is well written, and the subject is interesting. It presents a nice revision.  I only have minor comments/suggestions to help improving it:

On Table 1- I suggest ordering the species alphabetically. How did the authors classified bacteria as rhizospheric, symbiotic or even pathogenic- For instance, Burkholderia sp. it is well accepted the several roles this species can have, not only pathogenic.

Fig. 4- I suggest the authors try to do an integrated scheme with two ways, since the same initial HCA is in both, the first step is similar and there are some common intermediates.

 Fig. 5- This figure needs improvement. The colours chosen are very similar and for me it's difficult to see the differences between the tones of blue or green used. Furthermore, what is PhoR and how is it converted to PhoB? Is phosphate involved or any other factor, is there an activation? This should be presented in the scheme. For the pair SinR/SinI, the same comment: the latter arose from the first, but how? Then for ExpR in the presence of AHL, the same ExpR arises, a similar representation is used on the scheme.  AHL are the small green bullets? All this needs clarification.

Conclusion section- This should describe the major findings of the paper; however some species names and references are mentioned here for the first time: Neisseria; Staphylococcus aureus, Dehalococcoids mccartyi and Sulfolobus. If they are so important to be on the conclusions, describe them in the appropriate section of the manuscript. Avoid references in the conclusion section.

Minor corrections:

L88- Correct can be classified. Why the could?

L178- Correct could involve both the

L397- full name for D. dadantii

L418- full name for A. fabrum

I'm not a native speaker, but for me the language is of good quality.

Author Response

We thank reviewer 4 for his comments. We have answered point per point to the comments/suggestions in bold.

Comments and Suggestions for Authors

The paper is well written, and the subject is interesting. It presents a nice revision.  I only have minor comments/suggestions to help improving it:

On Table 1- I suggest ordering the species alphabetically. How did the authors classified bacteria as rhizospheric, symbiotic or even pathogenic- For instance, Burkholderia sp. it is well accepted the several roles this species can have, not only pathogenic.

We agree with reviewer 4 and we modified the table according to the request.

Fig. 4- I suggest the authors try to do an integrated scheme with two ways, since the same initial HCA is in both, the first step is similar and there are some common intermediates.

We agree with reviewer 4 and we modified the figure 4 according to the request.

 Fig. 5- This figure needs improvement. The colours chosen are very similar and for me it's difficult to see the differences between the tones of blue or green used. Furthermore, what is PhoR and how is it converted to PhoB? Is phosphate involved or any other factor, is there an activation? This should be presented in the scheme. For the pair SinR/SinI, the same comment: the latter arose from the first, but how? Then for ExpR in the presence of AHL, the same ExpR arises, a similar representation is used on the scheme.  AHL are the small green bullets? All this needs clarification.

We agree with reviewer 4 and we modified and simplified the figure 5 according to the request.

Conclusion section- This should describe the major findings of the paper; however some species names and references are mentioned here for the first time: Neisseria; Staphylococcus aureus, Dehalococcoids mccartyi and Sulfolobus. If they are so important to be on the conclusions, describe them in the appropriate section of the manuscript. Avoid references in the conclusion section.

We agree with reviewer 4 and we modified the text accordingly.

Minor corrections:

L88- Correct can be classified. Why the could?

We corrected the “could” (L. 89).

L178- Correct could involve both the

We corrected the “involve” (L. 178).

L397- full name for D. dadantii

We indicated the full name (L. 393).

L418- full name for A. fabrum

We indicated the full name (L.414).